# The Complex Emotion Expression Database: A validated stimulus set of trained actors

**Margaret S. Benda**, **K. Suzanne Scherf** *

Department of Psychology, Pennsylvania State University, University Park, PA, United States of America

* suzyscherf@psu.edu

## Abstract

The vast majority of empirical work investigating the mechanisms supporting the perception and recognition of facial expressions is focused on basic expressions. Less is known about the underlying mechanisms supporting the processing of *complex expressions*, which provide signals about emotions related to more nuanced social behavior and inner thoughts. Here, we introduce the Complex Emotion Expression Database (CEED), a digital stimulus set of 243 basic and 237 complex emotional facial expressions. The stimuli represent six basic expressions (angry, disgusted, fearful, happy, sad, and surprised) and nine complex expressions (affectionate, attracted, betrayed, brokenhearted, contemptuous, desirous, flirtatious, jealous, and lovesick) that were posed by Black and White formally trained, young adult actors. All images were validated by a minimum of 50 adults in a 4-alternative forced choice task. Only images for which $\geq$ 50% of raters endorsed the correct emotion label were included in the final database. This database will be an excellent resource for researchers interested in studying the developmental, behavioral, and neural mechanisms supporting the perception and recognition of complex emotion expressions.

**Data Availability Statement:** All relevant data are within the paper and its Supporting Information files. The images are publicly available at Databrary. com (http://doi.org/10.17910/b7.874).

The ability to visually perceive, interpret, and categorize emotional expressions from the face is a central component of social communication. Even in infancy, humans perceive and use facial expressions as social signals (e.g., [1]). By adulthood, we are experts at using emotional expressions to predict and guide behavior.

The vast majority of empirical work investigating the developmental, behavioral, and neural mechanisms supporting the perception and recognition of facial expressions is focused on *basic expressions* (e.g., [2–9]). These expressions provide signals about a specific set of universal emotions, including: anger, disgust, fear, happiness, sadness, and surprise [10]. Much less is known about the underlying mechanisms supporting the perception and identification of non-basic, *complex expressions*, which provide signals about emotions related to more nuanced social behavior and inner thoughts [11]. The developmental, behavioral, and neural mechanisms supporting the perception and recognition of complex expressions could be similar, different, or overlapping. There is some existing work suggesting that the developmental and behavioral mechanisms may be different [see 12–13].

To facilitate more research on the processing of complex emotion expressions, we developed and validated a database of digital photographs showing young adult actors making

**Funding:** This work was supported by a grant from the National Institute of Mental Health R01MH112573 (KSS; https://www.nimh.nih.gov/index.shtml). The funders had no role in study design, data collection and analysis, decision to publish, or preparation of the manuscript.

**Competing interests:** The authors have declared that no competing interests exist.

complex expressions. Here, we discus categories of complex expressions and explain why the database primarily consists of basic and *complex social sexual* expressions [12]. Next, we review limitations of existing databases that do feature complex expressions and show how this new database fills some of those gaps. Finally, we identify the primary goals and strategies motivating the development of this database as well as the validation procedure and results.

## Categories of complex emotional expressions

Based on the notion that expressions provide adaptive social signals that enable rapid appraisal and preparation to act [14], several investigators have suggested there are subcategories of complex expressions based on their signaling properties [11, 12, 15]. For example, there is precedent for distinguishing *complex cognitive* and *complex social expressions* (for review see [12]). Briefly, complex cognitive expressions (e.g., pensive) reflect inner thoughts (i.e., they do not necessarily result from interactions with people) and have low valence and arousal, while complex social expressions are elicited in specific social contexts and vary in arousal and valence (e.g., serious) [15]. Recently, we proposed additional organizational structure within the category of complex social expressions to include sub-categories, like *social self-conscious* (e.g., guilt, pride) and *social sexual* (e.g., desire, flirtatious) expressions [12]. We argued that these sub-categories reflect additional functional segregation of the signaling properties of the expressions. Self-conscious expressions facilitate behavioral adherence to moral standards and are evoked by self-reflection and self-evaluation. In contrast, social sexual expressions provide signals about the status of romantic and sexual relationships. For example, the eyebrow flash together with a smile is a signal of sexual interest [16].

## Existing emotional expression databases

There is a plethora of digital stimuli databases for researchers to access for researching questions about the perception and recognition of basic expressions. These databases largely include static photographs and vary by age of the expresser to include infants, children, adolescents, or adults [17, 18, 19, 20]. Most of the databases include both male and female expressers and some have racial/ethnic diversity (e.g., [21, 22, 23]). In contrast, there are many fewer stimulus database options for researchers interested in investigating the perception and/or recognition of complex expressions.

We have summarized all of the available stimulus databases that include some form of complex emotional expression in Table 1. What is evident is that when complex expressions are included in a stimulus database, there are not many categories of complex expressions to draw upon for study as a researcher. For example, several databases only include the single complex expression of contempt [21, 29, 36, 38]. This expression is not representative of all complex expressions and may not even represent a broad category of complex expressions very well. As a result, researchers cannot learn much about the perception and recognition of complex expressions more generally by investigating responses to this single expression.

Across these databases, the most frequently represented complex expressions are social self-conscious expressions, including pride and/or embarrassment [19, 20, 24, 26, 30, 35]. These stimuli are particularly useful for asking questions related to the perception and recognition of signals about adherence to moral standards that involve self-reflection and self-evaluation. For example, they would be particularly useful for addressing questions about clinical or at-risk populations that exhibit impairments in these abilities or about the developmental emergence of sensitivity to these expressions. However, if researchers are interested in other kinds of questions like whether there are age-related changes in sensitivity to social sexual expressions or

**Table 1. Existing emotion expression stimulus databases that include complex expressions.**

| Database | Paper | Complex Expressions | Actors as Models | Age of Models | Number of Models | Ethnicity of Models |
|---|---|---|---|---|---|---|
| ADFES | [24] | contempt, embarrassment, pride | No | 18–25 yrs | 20 + 2 authors | White |
| BINED | [25] | frustration, amusement | No | "Adults" | 256 | White, Latinx |
| BP-4D Spontaneous | [26] | embarrassment, pain | No | 18–29 yrs | 41 | Asian, Black, Hispanic, White |
| CAM Face-Voice Battery | [27] | appalled, appealing, confronted, distaste, empathic, exonerated, grave, guarded, insincere, intimate, lured, mortified, nostalgic, reassured, resentful, stern, subdued, subservient, uneasy, vibrant | Yes | "Adults" | Not provided | Not provided |
| Chicago | [28] | threatening | Some | 18–40 yrs | 158 | Black, White |
| CK + | [29] | contempt | Not provided | 18–50 yrs | 210 | Black, White, other |
| Dartmouth | [18] | content | No | 6–16 yrs | 80 | White |
| DuckEES | [19] | pride, embarrassment | Yes | 8–18 yrs | 36 | White, other |
| DynEmo | [30] | amusement, interest, irritation, worry | No | "undergraduates" | 43 | Not provided |
| Eu-Emotion | [20] | ashamed, bored, disappointed, excited, frustrated, hurt, interested, jealous, joking, kind, proud, sneaky, unfriendly, worried | Yes | 10–70 yrs | 19 | White, Black, Mixed |
| GEMEP-CS | [31] | amusement, anxiety, cold anger (irritation), despair, hot anger (rage), fear (panic), interest, joy (elation), pleasure (sensory), pride, relief, admiration, contempt, tenderness | Yes | 25–57 yrs | 10 | Not provided |
| JACFEE | [21] | contempt | Not provided | Not provided | 56 | Asian, White |
| McEwan Faces | [32] | compassionate, critical | Yes | "young actors," "mature actors" | 31 | White, Black, Asian |
| McGill Face Database | [33] | 92 expressions: affectionate, comforting, contemplative, desire, distrustful, embarrassed, flirtatious, friendly, imploring, jealous, playful, and sympathetic | Yes | 23–29 yrs | 2 | White |
| MPI | [34] | agree, aha, arrogance, bored, annoyed, confused, contempt, don't care, didn't hear, disagree, disbelief, don't know, don't understand, embarrassment, evasive, imagine, impressed, insecurity, compassion, not convinced, pain, annoyed, thinking, smiling, tired, doe-eyed | No | 20–30 yrs | 19 | Not provided |
| MSFDE | [35] | embarrassment, shame | Yes | "Young adults" | 24 | Asian, White, Black |
| Radboud | [36] | contempt | No | "Children," "Adults" | 49 | White |
| STOIC | [37] | pain | Yes | 20–45 yrs | 10 | Not provided |
| TFEID | [38] | contempt | Not provided | Not provided | 40 | Not provided |
| UT Dallas | [39] | boredom, disbelief, laughter, puzzlement | No | 18–41 yrs | 284 | White, Black, Asian, Hispanic, Other |

whether sensitivity to these expressions changes as a function of pubertal development or relationship status (i.e., single, committed romantic relationship), there are limited options.

The CAM Face-Voice Battery includes the expressions of appealing, empathic, and intimate [27]. The GEMEP database includes the complex social expressions of tenderness, interest, and pleasure [31]. The McGill Face Database [33] is composed of 93 different expressions that include basic, complex cognitive, and complex social expressions. While the McGill Database has an excellent range of complex expression stimuli, the limitation is that there are only two actors (one male, one female) who portray all the expressions, which restricts the generalizability of any findings. Specifically, there is a confound between the actor and the

expression such that it is impossible to discern whether differential responses are specific to the actor, expression, or the interaction between the two. It is important to have multiple actors portraying the same expression to prevent this confound. Finally, if researchers cobble together stimuli across multiple databases in a single experiment, they risk including methodological differences that were introduced in the creation of the stimuli across expressions as an additional confound. Given the limitations in the existing databases, there is a clear need for complex expression stimuli, especially those that portray expressions from the multiple subcategories of expressions.

## Complex Emotion Expression Database (CEED)

Here, we introduce a database of facial emotional expressions that we developed to address some of these limitations. The database includes both basic and complex social expressions portrayed by young adult actors with formal training and extensive performance experience. In particular, the complex social expressions represent the subcategory of social sexual expressions, including affectionate, attracted, betrayed, brokenhearted, contemptuous, desirous, flirtatious, jealous, and lovesick. We were especially interested in creating a database of these expressions because they are underrepresented in the literature and they are central to hypotheses that we are exploring in our own work. Previously, we predicted that emerging perceptual sensitivity to different subcategories of complex expressions may follow different developmental trajectories based on the relevance of each for accomplishing social developmental tasks of childhood and adolescence [12]. For example, we predicted that sensitivity to complex social sexual expressions would only emerge in adolescence and as a function of pubertal development as adolescents begin to explore and participate in romantic and sexual partnerships with peers, which is a primary social developmental task of adolescence [12, 40]. We are using these stimuli to address these questions.

The Complex Emotion Expression Database (CEED) includes 480 images of eight young adult actors creating six basic and nine complex social sexual emotional expressions. The actors are both female and male with some racial diversity. The images were independently rated by nearly 800 participants to validate the perception of the expression.

## Method

### Participants

**Models.** Eight professional actors were hired to pose basic and complex facial emotional expressions. The actors were young adults who ranged in age from 18–27 years ($M$ = 20.9 years, $SD$ = 3.1), and included four Black and four White individuals, and four males and four females (see Table 2). All of the actors had formal training in acting. They were all in either undergraduate or graduate level theater training programs and had extensive performance histories. The actors all provided written consent for their photos to be taken and used for research purposes. The individuals pictured in Figs 1 and 2 of this manuscript have provided written informed consent (as outlined in PLOS consent form) to publish their image alongside the manuscript.

**Raters.** A total of 870 people rated the images. However, it is possible that these are not all unique individuals because we did not preclude raters from participating in the multiple versions of the task that were available on Amazon's Mechanical Turk (MTurk). The raters ranged in age from 18–82 years old ($M$ = 34.5 years, $SD$ = 11.4). The self-reported gender-identity of the sample included 449 males, 416 females, 1 gender non-conforming individual, and 4 individuals who did not report a gender identity. We did not ask raters to report their racial or ethnic identity. Participants provided implied consent online before they rated the stimuli. These procedures were approved by the Institutional Review Board of Penn State University.

**Table 2. Actor demographics.**

| Actor ID | Age (Years) | Gender | Ethnicity | # of Images |
|---|---|---|---|---|
| 20yoWM1 | 20 | M | White | 81 |
| 18yoWM2 | 18 | M | White | 71 |
| 27yoBM1 | 27 | M | Black | 68 |
| 19yoBM2 | 19 | M | Black | 58 |
| 19yoWF1 | 19 | F | White | 74 |
| 21yoWF2 | 21 | F | White | 15 |
| 24yoBF1 | 24 | F | Black | 88 |
| 19yoBF2 | 19 | F | Black | 25 |
| Mean (SD) | 20.9 (3.1) | | | 60.0 (26.3) |

## Procedure

**Photographing the expressions.** Actors were invited to the Laboratory of Developmental Neuroscience to be photographed in individual sessions. Photographs were taken using a Fujifilm FinePix S4200 digital camera with a 24 mm lens. The photographs were taken in a quiet room with the actors sitting on a chair in front of a white wall. Lighting was standardized across sessions, which included overhead fluorescent lights, an umbrella reflector, and an 85-watt spotlight on the actor's face. The camera was stabilized on a tripod approximately five feet away from the actor. The zoom function was used to focus in on the actor's shoulders, neck, and head.

Images were acquired of six basic (angry, disgusted, fearful, happy, sad, surprised) and nine socially complex expressions (affectionate, attracted, betrayed, brokenhearted, contemptuous, desirous, flirtatious, jealous, lovesick). Emotional expressions were photographed in separate blocks (i.e., all angry expressions, all contempt expressions). Prior to photographing an expression, the researchers provided the actors with a definition and image of the target emotional expression. The actors were instructed to express the specific emotional expression using a method acting approach. The researchers also provided a series of example scenarios that would elicit the emotional expression. The researchers gave the actors time to invoke the emotion and asked the actors to let the researcher know when they were ready to generate the emotional expression to be photographed. The actors produced the expression as the researcher took multiple photographs of the actor.

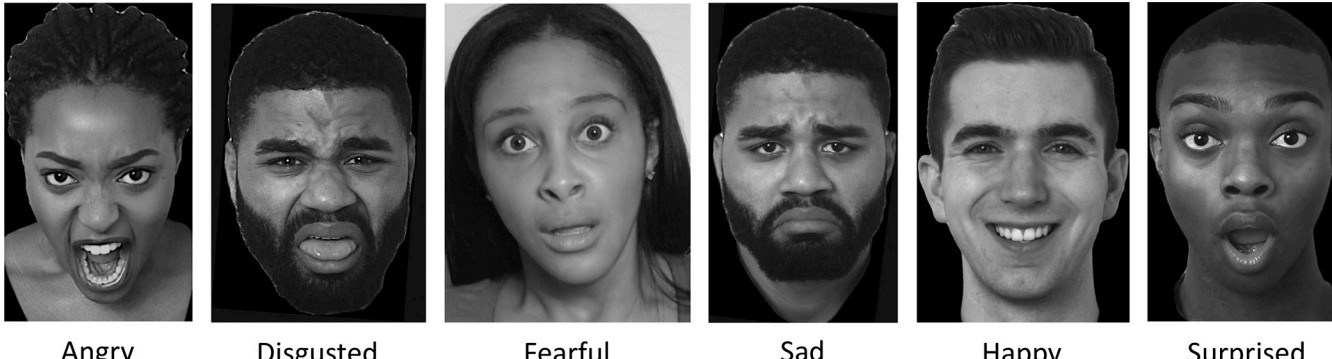

| Angry | Disgusted | Fearful | Sad | Happy | Surprised |

**Fig 1. Highly rated basic expression images in CEED.** Ratings were measured in a 4AFC task and computed as endorsement scores. The scores reflect the percent of total raters who endorsed the target label for the expression (i.e., picked the label "angry" for an angry expression). Only images with endorsement scores ≥ 50% are included in the database. These images have scores that range from 87–94.5%.

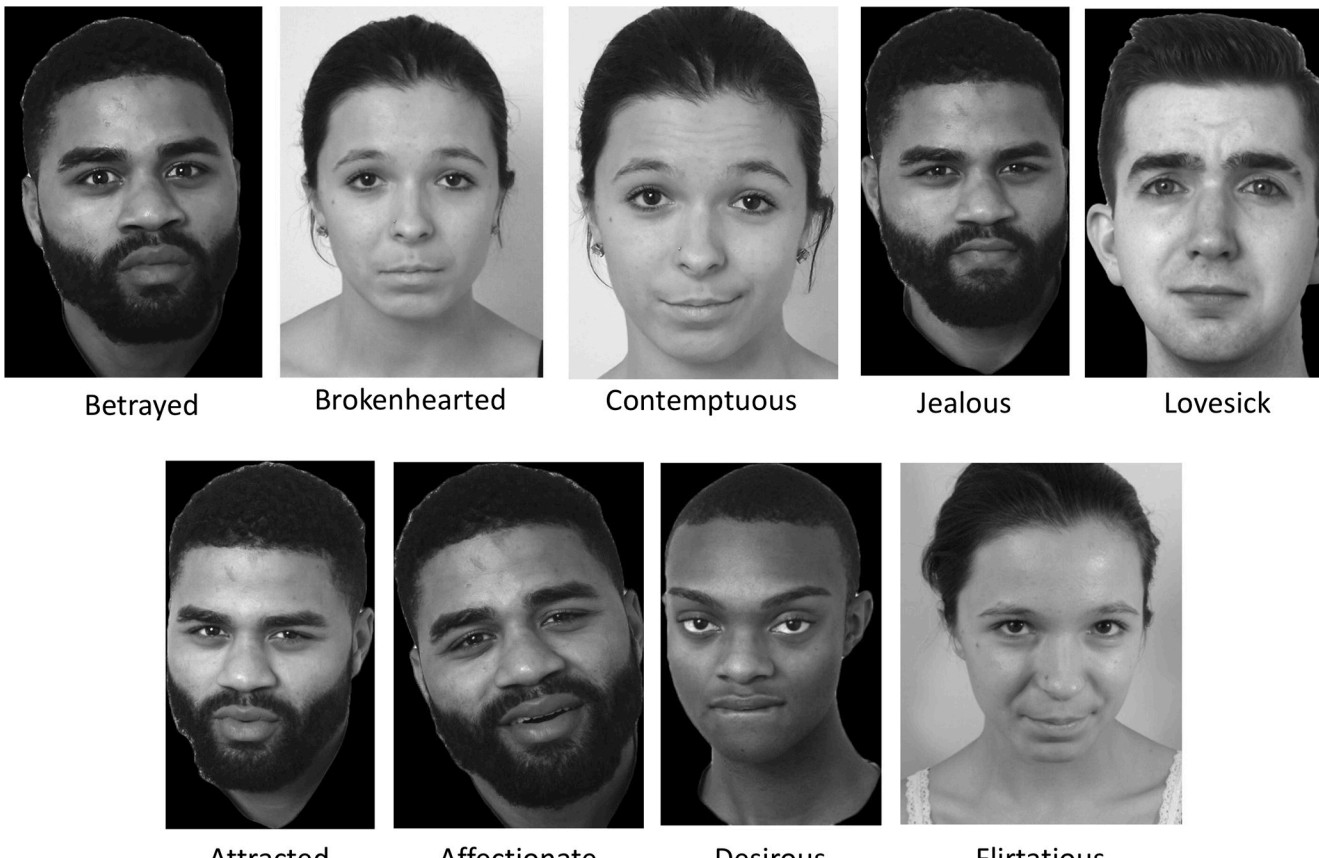

**Fig 2. Highly rated complex expression images in CEED.** Ratings were measured in a 4AFC task and computed as endorsement scores. The scores reflect the percent of total raters who endorsed the target label for the expression (i.e., picked the label "jealous" for a jealous expression). Only images with endorsement scores ≥ 50% are included in the database. These images have scores that range from 64.5–81.5%.

**Image processing.** Images that were blurry or that were determined by the researchers not to express the target emotion were excluded from further processing. Images that were selected for additional validation were cropped in Photoshop to remove much of the background scene. This procedure was not fully standardized so some of the images were cropped to only include the head and neck of the actor. Other images include the shoulders of the actor and some of the wall behind the actor. All of the images are greyscaled and standardized to a resolution of 300 dpi and a width of 4 inches. The length of the images varies slightly.

**Validating the expression stimuli.** To measure the extent to which the actor was successfully able to produce the emotional expression they were asked to express, the images were externally rated. Participants were recruited from MTurk to rate the expressions of the images. This was executed in five versions of the validation task on MTurk, with each iteration including different images. As a result, different participants rated different photographs. However, we did not limit rater participation to just one of the validation tasks, so it is possible that some participants rated multiple images. A total of 719 images (296 basic and 423 complex) were rated on MTurk. Every image was rated by at least 50 people (number of raters per image $M = 100.1$, $SD = 84.7$).

The validation task was a 4-alternative forced choice task (4AFC). Raters were presented with a single image and four emotion expression labels simultaneously. They were instructed to pick the best label that described the expressed emotion. Raters had an unlimited amount of

time to respond and the image stayed on the screen until the rater provided a response. The order of the images was randomized within each task for each participant.

Critically, the alternative label choices were systematically controlled. First, the alternative label options were always within valence, which prevented participants from easily eliminating potential labels on the basis of valence. For example, if a participant was shown a negatively valenced emotion expression (e.g., angry), then all the labels described negatively valenced expressions (e.g., angry, disgusted, arrogant, sarcastic). Second, the labels always included descriptions of both basic and complex emotion expressions. This was essential to identify the specificity of the perception of the expression. For the images featuring basic expressions, the labels included the target expression (e.g., angry), an alternative basic expression (e.g., disgust), and two alternative complex expressions (e.g., arrogant, sarcastic). Similarly, for the images featuring complex expressions, the labels included the target expression (e.g., jealous), an alternative complex expression (e.g., despondent), and two alternative basic expressions (e.g., sad, fearful). Third, the labels describing complex expressions were selected from the labels used in the Revised Reading the Mind in the Eyes Test (RMET) [41]. The RMET is a commonly used measure of emotion expression perception that includes complex expressions and presents participants with a similar 4AFC paradigm. The answer choices for each expression are provided in S1 Table.

**Data analysis.** The primary dependent variable was accuracy, the ability to identify the target label for the emotional expression represented in the image. Raters did not have to provide complete data to be included in the analyses. We included partial data in the analyses. Missing data were not interpolated. To identify unengaged participants, we excluded those who scored at or below chance level performance. To do so, we computed total mean accuracy across all trials for each participant. Raters whose total accuracy was $\leq 25\%$ were excluded from the analysis (n = 74).

To determine the external validity of the emotional expression represented in each image, we computed *endorsement scores*. The endorsement score was defined as the percent of total raters who endorsed the target label for the expression (i.e., picked the label "angry" for an angry expression). This is essentially an accuracy score if one considers the target label the "correct" label. Therefore, the frequency with which participants picked an alternative label reflects the "error" on a given trial.

The endorsement scores for all images in the database are provided in S2 Table. We used the endorsement scores to further down select the items in the database so that only items with good external validity are provided. We include all images with endorsement scores $\geq 50\%$, which indicates that an absolute minimum of 25 people (50% of 50 raters) endorsed the expression displayed in the image to be represented by the target label when presented with three alternative labels that were of similar valence. In addition, we analyzed the error on each item to identify the proportion of raters who chose each of the alternative labels. For example, researchers can determine whether raters are likely to misidentify an expression (e.g., anger) consistently with the label for another expression (e.g., betrayed) or not. Together with the endorsement scores, this error information helps to define the relative specificity of these scores for each stimulus.

## Results

The final set of raters who provided data for the analyses of the endorsement scores of the expression images included 796 individuals. They ranged in age from 18–82 years old ($M$ = 34.8 years, $SD$ = 11.6). The self-reported gender-identity of the raters included 403 males, 388 females, 1 individual who identified as gender non-conforming, and 4 individuals whose gender was not provided.

Participants rated a total of 719 images. From these images, a total of 480 (243 basic and 237 complex) images (66.8%) elicited endorsement scores $\geq$ 50% and are included in the database. In S2 Table, the images are cataloged with endorsement scores as well as information about the raters (number, gender) and the frequency of responses to the alternative labels.

## Endorsement scores for images representing basic expressions

Among the 243 images representing basic emotion expressions in the database, 43 are of angry, 26 are of disgusted, 48 are of fearful, 49 are of happy, 33 are of sad, and 44 are of surprised expressions. Fig 1 illustrates highly endorsed images representing each of these six basic expressions. Endorsement scores for images representing basic expressions ranged from 50.0–95.3% ($M$ = 75.0%, $SD$ = 11.0%) (see Table 3). The images representing surprise expressions generated the highest endorsement score ($M$ = 79.3%, $SD$ = 12.8%) and the images representing disgusted expressions generated the lowest endorsement score ($M$ = 70.6%, $SD$ = 9.5%). All eight actors contributed images representing the expressions of fearful, happy, and surprise. Seven actors contributed images representing the expression of anger and six actors contributed images representing the expressions of disgusted and sad.

These results reveal consistent endorsement of the target label for the expressions in the images. In addition, there was also specificity in the perception of the expressions. For example, across all the images in which basic expressions were exhibited, the alternative basic ($M$ = 6.3%, SD = 7.5%) and alternative complex (M = 9.4%, SD = 7.5%) labels were rarely endorsed (see Table 3). The only set of images in which there was some willingness to endorse an alternative label was for the expression of surprise. On average 11.7% (SD = 12.3) of raters chose the alternative basic label "happy." Also, it is important to note that although the basic expression labels were repeated more frequently than were alternative labels (because there are fewer to choose from within valence), this did not bias raters' selection of alternative labels. S2 Table shows that for all basic expressions, the most common alternative label picked was a complex (not basic) expression label. Similarly, for complex expressions, the most common alternative label picked was also a complex label, even though there were two basic label options.

## Endorsement scores for images representing complex expressions

Among the 237 images representing complex expressions in the database, 36 are of affectionate, 19 are of attracted, 20 are of betrayed, 36 are of brokenhearted, 19 are of contemptuous, 46 are of desirous, 22 are of flirtatious, 9 are of jealous, and 30 are of lovesick expressions. Fig 2

**Table 3. Endorsement scores for basic images.**

| Expression | # of Images | Mean | Range | Alternative Basic | Alternative Complex |
|---|---|---|---|---|---|
| Angry | 43 | 76.6 (10.8) | 52.0–93.0 | 6.5 (7.9) | 8.5 (5.9) |
| Disgusted | 26 | 70.6 (9.5) | 51.8–87.0 | 8.0 (6.6) | 10.7 (5.4) |
| Fearful | 48 | 76.5 (10.6) | 52.4–93.0 | 3.5 (2.5) | 10.0 (6.6) |
| Happy | 49 | 71.2 (9.8) | 50.8–88.3 | 3.9 (3.6) | 12.4 (10.3) |
| Sad | 33 | 74.1 (10.0) | 53.5–90.6 | 5.2 (4.3) | 10.4 (8.0) |
| Surprised | 44 | 79.3 (12.8) | 50.0–95.3 | 11.7 (12.3) | 4.5 (3.5) |
| **Basic** | **243** | **75.0 (11.0)** | **50.0–95.3** | **6.3 (7.5)** | **9.4 (7.5)** |

Contents represent Mean (SD). Images are included in final dataset if mean endorsement score $\geq$ 50%.

**Table 4. Endorsement scores for complex images.**

| Expression | # of Images | Mean Score | Range | Alternative Complex | Alternative Basic |
|---|---|---|---|---|---|
| Affectionate[a] | 36 | 62.0 (6.5) | 50.8–75.5 | 16.0 (8.1) | 6.1 (3.0) |
| Attracted | 19 | 63.9 (8.2) | 50.0–81.5 | 17.3 (8.0) | 9.4 (8.9) |
| Betrayed | 20 | 57.2 (5.7) | 50.0–64.5 | 16.6 (6.1) | 13.1 (6.1) |
| Brokenhearted | 36 | 63.5 (7.9) | 50.0–81.2 | 19.7 (7.8) | 8.4 (6.4) |
| Contemptuous | 19 | 61.9 (7.0) | 51.6–75.4 | 17.3 (7.3) | 10.4 (7.7) |
| Desirous[a] | 46 | 65.3 (9.2) | 50.8–79.6 | 15.3 (10.0) | 4.2 (2.8) |
| Flirtatious | 22 | 62.0 (7.3) | 50.9–74.4 | 12.8 (7.1) | 12.6 (11.6) |
| Jealous | 9 | 60.3 (4.9) | 51.5–67.9 | 25.5 (4.1) | 7.1 (4.5) |
| Lovesick | 30 | 60.6 (6.5) | 50.9–77.4 | 8.0 (3.6) | 15.7 (8.4) |
| **Complex** | **237** | **62.4 (7.7)** | **50.0–81.5** | **15.7 (8.7)** | **10.0 (8.1)** |

Contents represent Mean (SD). Images are included in final dataset if mean endorsement score ≥ 50%.

[a]Raters were shown one alternative basic label and two alternative complex labels in addition to the correct complex label.

illustrates highly endorsed images representing each of these complex expressions. Endorsement scores for images representing complex expressions ranged from 50.0%– 81.5% ($M$ = 62.4%, $SD$ = 7.7%) (see Table 4). The images representing the expression of desirous generated the highest endorsement score ($M$ = 65.3%, $SD$ = 9.2%), whereas images representing the expression of betrayed generated the lowest endorsement score ($M$ = 57.2%, $SD$ = 5.7%). All eight actors contributed images representing the expressions of desirous and brokenhearted. Seven actors contributed images representing the expression of flirtatious; six actors contributed images representing the expressions of attracted, betrayed, and lovesick; five actors contributed images representing the expression of contemptuous; and three actors contributed images representing the expression of jealousy.

These results reveal consistent, but slightly lower, endorsement of the target labels for the complex expressions in the images. As with the basic expression images, there was also specificity in the perception of the expressions. For example, across all the images in which complex expressions were exhibited, the alternative complex ($M$ = 15.7%, $SD$ = 8.7%) and alternative basic ($M$ = 10.0%, $SD$ = 8.1%) labels were rarely endorsed (see Table 4). The images in which there was the most willingness to endorse an alternative label was for the expression of jealous. On average 25.5% ($SD$ = 4.1) of participants chose the alternative complex label "despondent". The images with the expressions of affectionate, attracted, betrayed, brokenhearted, contemptuous, desirous, and jealous were most often mistaken for another complex expression. However, lovesick was most often mistaken for one of the alternative basic expressions, either fearful or disgusted.

The full set of images that were endorsed ≥ 50% are available for download for research purposes at Databrary (http://doi.org/10.17910/b7.874).

## Discussion

Here, we describe a database of facial emotional expressions posed by formally trained, young adult actors. There are both basic (six) and complex social (nine) expressions, with a particular emphasis on *social sexual* (e.g., desire, flirtatious) expressions [12]. All of the individual stimuli were validated by a minimum of 50 adult raters and have endorsement scores at or over 50% in a 4AFC paradigm. There are 480 images in the final database: 243 images represent basic expressions and 237 images represent complex expressions.

We provided images that were minimally edited to reflect the way they were rated on MTurk and that met a low threshold of endorsement to provide researchers with flexibility in determining how to work with the stimuli. Importantly, researchers may want to take additional steps with the stimuli to process or control multiple physical characteristics of the images. For example, in our own work, we match images for luminance, use a higher endorsement threshold to down select stimuli for our tasks, and crop the images so that only the head and neck is visible in each image. Researchers may also consider cropping out hair.

There are some limitations to consider regarding these stimuli. First, a central goal of designing these stimuli was to include multiple actors who posed the same expressions to avoid a confound between actor and expression. However, not all actors generated exemplar expressions that surpassed the endorsement threshold. As a result, there is variability in the number of images each actor contributed to the final stimulus set. Second, jealous was a particularly difficult expression to either create or distinguish from other expressions. The stimulus set only contains nine images of the jealous expression (all male) that met the minimum endorsement criterion. Third, the complex expressions in the database are all of the social sexual category because of our own work and hypotheses. It will be essential that similar databases with social self-conscious and other kinds of social complex or cognitive complex expressions be created and validated.

It is important to note that the focus of this study was to validate the external validity of the stimuli as complex emotional expressions. Our goal was to amass ratings from a large, representative sample of adults to help validate the stimuli. As a result, we tested people who self-identified as male, female, and non-binary genders, and who self-reported being 18–82 years of age. Importantly, we did not design this study as an empirical investigation of the effects of these participant factors (e.g., gender, age) on the perception of the expressions. This requires careful balancing of groups for confounding factors and methodological procedures (e.g., screening for psychopathology which can vary by gender). However, these are exactly the kinds of questions we hope researchers will pursue using the CEED stimuli. Specifically, are there effects of gender and/or age of the observer and/or stimulus face on the perception of basic/complex expressions?

By making CEED available to researchers, this will support the study of the developmental, behavioral, and neural basis of facial emotion expression processing. This database will enable researchers to study the perception and recognition of complex social expressions, particularly social sexual expressions, which have previously been underrepresented in the literature. Using the basic and complex stimuli available in CEED, future researchers can develop a better understanding of how perception of basic and complex expressions differ, how sensitivity to certain emotion expressions develops over the lifespan, and how social contexts can impact the perception of emotion expressions. The focus on complex expressions in this database will begin to fill in the gaps in the current literature and allow for nuanced questions about emotion expression to be addressed.

## Supporting information

**S1 Table. Emotion labels.** This table includes the emotion label answer choices presented to the MTurk raters in the validation task. It lists the four emotion labels presented for each of the 15 types of facial emotion expression. This includes the target label expression, the alternative basic expression(s), and the alternative complex expression(s).
(DOCX)

**S2 Table. Image ratings.** Image ratings for all stimuli included in CEED. Includes information regarding the number and gender of raters, the answer choices provided with the image, and

the percentage of raters who chose each answer choice. The images are listed by expression type, actor, and file name.
(XLSX)

## Acknowledgments

We are grateful to the actors for making this research possible.

## Author Contributions

**Conceptualization:** K. Suzanne Scherf.

**Formal analysis:** Margaret S. Benda, K. Suzanne Scherf.

**Funding acquisition:** K. Suzanne Scherf.

**Writing – original draft:** Margaret S. Benda, K. Suzanne Scherf.

**Writing – review & editing:** Margaret S. Benda, K. Suzanne Scherf.

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
