## [Decision Letter · Decision Letter 0]

30 Oct 2019

PONE-D-19-25816

The Complex Emotion Expression Database: A validated stimulus set of trained actors.

PLOS ONE

Dear Dr Scherf,

Thank you for submitting your manuscript to PLOS ONE. After careful consideration, we feel that it has merit but does not fully meet PLOS ONE’s publication criteria as it currently stands. Therefore, we invite you to submit a revised version of the manuscript that addresses the points raised during the review process.

We would appreciate receiving your revised manuscript by Dec 14 2019 11:59PM. To enhance the reproducibility of your results, we recommend that if applicable you deposit your laboratory protocols in protocols.io, where a protocol can be assigned its own identifier (DOI) such that it can be cited independently in the future. For instructions see: http://journals.plos.org/plosone/s/submission-guidelines#loc-laboratory-protocols

We look forward to receiving your revised manuscript.

Kind regards,

Zezhi Li, Ph.D., M.D.

Academic Editor

PLOS ONE

Journal Requirements:

2. We note that Figures 1 and 2 includes an image of a [patient / participant / in the study]. 

Additional Editor Comments:

There are some concerns of two reviewers which should be addressed before we can consider its potential publication.

Reviewers' comments:

Reviewer's Responses to Questions

**Comments to the Author**

1. Is the manuscript technically sound, and do the data support the conclusions?

Reviewer #1: Yes

Reviewer #2: Yes

2. Has the statistical analysis been performed appropriately and rigorously? 

Reviewer #1: Yes

Reviewer #2: Yes

3. Have the authors made all data underlying the findings in their manuscript fully available?

Reviewer #1: Yes

Reviewer #2: Yes

4. Is the manuscript presented in an intelligible fashion and written in standard English?

Reviewer #1: Yes

Reviewer #2: No

5. Review Comments to the Author

Reviewer #1: Comment on PONE-D-19-25816

The paper entitled “The Complex Emotion Expression Database: A validated stimulus set of trained actors” By Margaret S. Benda et al introduced a Complex Emotion Expression Database (CEED), a digital stimulus set of 243 basic and 237 complex emotional facial expressions. Images were validated by around 50 adults in a 4-alternative forced choice task and shown in tables with those have higher than 50% raters endorsement and correct emotion labels.

Overall, the study is well designed. Results are well described. The authors have good discussions about their findings.

However, the following concerns need to be addressed:

All images were performed by young adult actors with ages ranging from 18 – 27 years old, while the raters are from 18 – 82 years old. It is possible that older people are more developed in their cognitive skills thus have different opinions (compared with young raters) towards the concept of the same expression. The author may want to discuss this point.

In the method section, the authors indicated that “To determine the specificity of the endorsement scores, we also computed the frequency with which raters picked the alternative labels.” Please introduce more details about how the frequency the raters picked the alternative labels affects the endorsement scores.

Do genders have effects on the judges of “correct expressions”?

The authors give several alternative labels for the same expression. It is possible that in some expression group the labels have quite similar meanings which make it difficult to judge correctly, therefore the expression has a low endorsement score. The author may want to discuss whether the types of alternative labels given affect the endorsement scores.

Other minor correction:

Please use justify text for the manuscript.

Reviewer #2: I am very happy to have the opportunity to review this article. Overall, I feel that it is not suitable to publish this paper on our magazine of high level. The main shortcoming is the introduction part. The author simply say that the specific mechanism of research on face recognition is unknown. In fact, there are still many positive results on mechanisms such as Neural pathway, biochemical mechanism etal. At the same time, the author does not explain the complex emotions in a more reasonable way. It is better to have a hypothesis instead of just comparing it with other face databases while this article also missed the introduction of the EKMan expression database which is recognized as a comprehensive expression library.

6. PLOS authors have the option to publish the peer review history of their article (what does this mean?). If published, this will include your full peer review and any attached files.

Reviewer #1: No

Reviewer #2: No

---

## [Author Response · Author response to Decision Letter 0]

13 Dec 2019

Response to Reviewers:

In general, the Reviewers and Editor were positive about the study, rating it as technically sound with rigorous statistical analyses and publicly available data and stimuli. The Editor noted that this study “has merit.” Reviewer 1 indicated that the study “is well designed,” the “results are well described,” and that we provided “good discussions about the findings.” 

Reviewer 1’s concerns were focused on requests for clarification and for additional information about how characteristics of the raters may have influenced the results of the validation study and how the endorsement scores were computed. Reviewer 2 requested additional information about the nature of complex emotional expressions. We have addressed each of these concerns below with detailed responses. We have also corrected all typographical errors and formatted the manuscript to be compliant with PLoS One requirements. We think the paper is much improved as a result.

Reviewer 1:

Although the stimuli included individuals ages 18-27 years, the raters included individuals ages 18-82 years. Is it possible that there are age-related effects in the perception of these expressions? The authors may want to discuss this point.

We agree with the Reviewer that this is an interesting possibility and exactly the kind of question we hope researchers will investigate using the CEED stimuli. Unfortunately, because this is a validation study of the stimuli; it was not designed to address this question empirically. For example, to investigate age-related effects on the perception of the stimuli, we would need to design the study ahead of time to ensure that there are equal numbers of male and female participants in the relevant age groups and we would have to clearly define the age groups a priori based on hypotheses. This study was not designed in this way.

However, we think this point reflects a concern on the part of the Reviewer about whether age-related effects on perception of the stimuli could affect the results of the validation study. In other words, if older raters had consistently higher endorsement scores on the stimuli, did more stimuli reach the 50% threshold for inclusion in the database? We agree that this is an important issue to address to the best of our abilities. Therefore, we looked at the mean endorsement scores for older and younger participants (defined by median split on age � 31 years) for several exemplar stimuli in each expression, including the highest-, lowest- and a middle-rated images. Across these images/expressions, there was no clear pattern in which either the younger or older participants consistently rated the images higher/lower. Therefore, we are confident that there is no consistent age-related influence on the endorsement ratings of these expressions that might impact the results of the validation study and the specific set of images that are included in CEED.

Given the importance of this point, we have added a paragraph in the Discussion that addresses these issues and recommends that this work be done in the future using the CEED stimuli. It reads (p. 18-19):

“It is important to note that the focus of this study was to validate the external validity of the stimuli as complex emotional expressions. Our goal was to amass ratings from a large, representative sample of adults to help validate the stimuli. As a result, we tested people who self-identified as male, female, and non-binary genders, and who self-reported being 18-82 years of age. Importantly, we did not design this study as an empirical investigation of the effects of these participant factors (gender, age) on the perception of the expressions. This requires careful balancing of groups for confounding factors and methodological procedures (e.g., screening for psychopathology which can vary by gender). However, these are exactly the kinds of questions we hope researchers will pursue using the CEED stimuli. Specifically, are there effects of gender and/or age of the observer and/or stimulus face on the perception of basic/complex expressions?”

Do genders have effects on the judges of “correct expressions”?

This is also a very interesting possibility that we did not design the validation study to investigate. When we design studies to investigate the potential effects of sex-differences on face processing, we are very careful to screen participants for ongoing symptoms or personal and family history of psychopathology, which can impact face processing and which is differentially represented across sex (see Scherf et al., 2017). Also, we make sure that there were no age differences across male and female participants. Neither of these critical methodological controls was implemented in this validation study. Therefore, although we know that 51% of the raters of the final set of stimuli self-identified as male and 49% self-identified as female, we do not have empirical data about sex differences in the perception of the CEED stimuli. To address the importance of this issue, we also mention the need to investigate the potential role of sex differences on the perception of these stimuli in the Discussion (p.18-19).

Please introduce more details about how the frequency the raters picked the alternative labels affects the endorsement scores. Researchers can determine whether raters confuse the expression more other another complex expression or another basic expression. 

Thank you for the suggestion. Recall that the endorsement score was defined as the percent of total raters who endorsed the target label for the expression (i.e., picked the label “angry” for an angry expression). 

We have addressed this concern and followed the recommendation in two ways. First, in the Methods section (p. 13), we have now clarified:

“This [the endorsement score] is essentially an accuracy score if one considers the target label the “correct” label. Therefore, the frequency with which participants pick an alternative label reflects the “error” on a given trial.

and

“In addition, we analyzed the error on each item to identify the proportion of raters who chose each of the alternative labels. For example, researchers can determine whether raters are likely to misidentify an expression (e.g., anger) consistently with the label for another expression (e.g., betrayed) or not. Together with the endorsement scores, this error information helps to define the relative specificity of these scores for each stimulus.”

We also highlighted findings from the recommended analysis in the Results section (p.15). 

“Also, it is important to note that although the basic expression labels were repeated more frequently as alternative labels (because there are fewer to choose from within valence), this did not bias raters’ selection of alternative labels. Supplementary Table 2 shows that for all basic expressions, the most common alternative label picked was a complex (not basic) expression label. Similarly, for complex expressions, the most common alternative label picked was also a complex label, even though there were two basic label options.

The authors may want to discuss whether the types of alternative labels given affect the endorsement scores.

To address this concern, we have provided more information about how we generated the alternative labels, particularly for the complex expressions. This is described in the Methods section on p. 12. It reads:

“The labels describing complex expressions were selected from the labels used in the Revised Reading the Mind in the Eyes Test (RMET) [40]. The RMET is a commonly used measure of emotion expression perception that includes complex expressions and presents participants with a similar 4-alternative forced choice paradigm.”

We purposefully avoided picking words that were semantically similar (e.g., fearful, terrified) to the target label.

Reviewer 2:

The main shortcoming is the introduction part. The author simply say that the specific mechanism of research on face recognition is unknown. In fact, there are still many positive results on mechanisms such as Neural pathway, biochemical mechanism etal

We think this concern reflects a misreading of our paper. Our reference to “mechanism” in paragraph 2 of the Introduction explains that the vast majority of existing work in the current literature is 

“investigating the developmental, behavioral, and neural mechanisms supporting the perception and recognition of facial expressions is focused on basic expressions.”

This is important because the mechanisms could be different, similar, or overlapping for perceiving and recognizing complex expressions. To make this point explicitly clear, we added the following sentence to the end of that paragraph (p.3):

“The developmental, behavioral, and neural mechanisms supporting the perception and recognition of complex expressions could be similar, different or overlapping. There is some existing work suggesting that the developmental and behavioral mechanisms may be different [see 12-13].”

The author does not explain the complex emotions in a more reasonable way. It is better to have a hypothesis instead of just comparing it with other face databases.

We would like to point out that we provided a clear definition for complex expressions in the Introduction of the manuscript on p. 3. It explains that complex expressions “provide signals about emotions related to more nuanced social behavior and inner thoughts” and comes from Baron-Cohen and colleagues (1997). We cited Garcia & Scherf (2015) because we review this paper together with additional evidence supporting the distinction between basic and complex emotions/expressions. Also, there is a section of the Introduction devoted to explaining complex social expressions under the heading – “Categories of complex emotional expressions” (p. 4). In this section, we describe categories of social complex expressions and the capacity (or lack thereof) of existing databases to be used to address scientific questions about the underlying developmental, behavioral and neural mechanisms supporting the processing of complex expressions. This is critical for demonstrating the critical need for a database of stimuli like CEED.

It is also important to note that this is a validation study to assess the external validity of the stimuli, not a study designed to investigate hypothesis-driven questions about the perception of these stimuli. We hope researchers will use the stimuli for exactly that purpose.

This article also missed the introduction of the EKMan expression database which is recognized as a comprehensive expression library.

Again, we think this is a misreading of our paper. We reference the Ekman & Friesen (1971) paper in which the original face stimuli were presented in the second paragraph of the Introduction (p. 3). It is reference number 10. Importantly, this database only includes basic expressions (happy, angry, disgusted, fearful, surprised, sad). This is why this paper is not listed in Table 1, which only lists databases that include complex expressions.

---

## [Decision Letter · Decision Letter 1]

13 Jan 2020

The Complex Emotion Expression Database: A validated stimulus set of trained actors.

PONE-D-19-25816R1

Dear Dr. Scherf,

We are pleased to inform you that your manuscript has been judged scientifically suitable for publication and will be formally accepted for publication once it complies with all outstanding technical requirements.

With kind regards,

Zezhi Li, Ph.D., M.D.

Academic Editor

PLOS ONE

Additional Editor Comments (optional):

Reviewers' comments:

Reviewer's Responses to Questions

**Comments to the Author**

1. If the authors have adequately addressed your comments raised in a previous round of review and you feel that this manuscript is now acceptable for publication, you may indicate that here to bypass the “Comments to the Author” section, enter your conflict of interest statement in the “Confidential to Editor” section, and submit your "Accept" recommendation.

Reviewer #1: All comments have been addressed

2. Is the manuscript technically sound, and do the data support the conclusions?

Reviewer #1: Yes

3. Has the statistical analysis been performed appropriately and rigorously? 

Reviewer #1: Yes

4. Have the authors made all data underlying the findings in their manuscript fully available?

Reviewer #1: Yes

5. Is the manuscript presented in an intelligible fashion and written in standard English?

Reviewer #1: Yes

6. Review Comments to the Author

Reviewer #1: (No Response)

7. PLOS authors have the option to publish the peer review history of their article (what does this mean?). If published, this will include your full peer review and any attached files.

Reviewer #1: No

---

## [Editor Report · Acceptance letter]

23 Jan 2020

PONE-D-19-25816R1 

The Complex Emotion Expression Database: A validated stimulus set of trained actors. 

Dear Dr. Scherf:

I am pleased to inform you that your manuscript has been deemed suitable for publication in PLOS ONE. Congratulations! Your manuscript is now with our production department. 

With kind regards,

on behalf of

Dr. Zezhi Li 

Academic Editor

PLOS ONE